# Evaluation of DNA Methylation Array for Glioma Tumor Profiling and Description of a Novel Epi-Signature to Distinguish IDH1/IDH2 Mutant and Wild-Type Tumors

**DOI:** 10.3390/genes13112075

**Published:** 2022-11-09

**Authors:** Laila C. Schenkel, Joseph Mathew, Hal Hirte, John Provias, Guillaume Paré, Michael Chong, Daria Grafodatskaya, Elizabeth McCready

**Affiliations:** 1Faculty of Health Sciences, Department of Pathology and Molecular Medicine, McMaster University, 1280 Main Street West, Hamilton, ON L8S 4K1, Canada; 2Faculty of Health Sciences, Department of Oncology, McMaster University, 699 Concession Street, Hamilton, ON L8V 5C2, Canada; 3Hamilton Regional Laboratory Medicine Program, Hamilton Health Sciences and St. Joseph’s Healthcare Hamilton, 50 Charlton Avenue East, Hamilton, ON L8N 4A6, Canada; 4Population Health Research Institute, 237 Barton Street East, Hamilton, ON L8L 2X2, Canada; 5Faculty of Health Sciences, Department of Biochemistry and Biomedical Sciences, McMaster University, Hamilton, ON L8S 4K1, Canada

**Keywords:** glioma, whole-genome methylation array, IDH1/IDH2 epi-signature, 1p19q co-deletion

## Abstract

Molecular biomarkers, such as *IDH1/IDH2* mutations and 1p19q co-deletion, are included in the histopathological and clinical criteria currently used to diagnose and classify gliomas. *IDH1/IDH2* mutation is a common feature of gliomas and is associated with a glioma-CpG island methylator phenotype (CIMP). Aberrant genomic methylation patterns can also be used to extrapolate information about copy number variation in a tumor. This project’s goal was to assess the feasibility of DNA methylation array for the simultaneous detection of glioma biomarkers as a more effective testing strategy compared to existing single analyte tests. Methods: Whole-genome methylation array (WGMA) testing was performed using 48 glioma DNA samples to detect methylation aberrations and chromosomal gains and losses. The analyzed samples include 39 tumors in the discovery cohort and 9 tumors in the replication cohort. Methylation profiles for each sample were correlated with IDH1 p.R132G mutation, immunohistochemistry (IHC), and previous 1p19q clinical testing to assess the sensitivity and specificity of the WGMA assay for the detection of these variants. Results: We developed a DNA methylation signature to specifically distinguish a *IDH1*/*IDH2* mutant tumor from normal samples. This signature is composed of 11 CpG sites that were significantly hypermethylated in the *IDH1/IDH2* mutant group. Copy number analysis using WGMA data was able to identify five of five positive samples for 1p19q co-deletion and was concordant for all negative samples. Conclusions: The DNA methylation signature presented here has the potential to refine the utility of WGMA to predict *IDH1/IDH2* mutation status of gliomas, thus improving diagnostic yield and efficiency of laboratory testing compared to single analyte IDH1/IDH2 or 1p19q tests.

## 1. Introduction

Glioma is one of the most common primary brain tumors representing approximately 25% of all primary brain and other central nervous system tumors and 80% of malignant tumors [1]. Prior to the 2016 publication of the WHO classification of central nervous system tumors [2], classification of glioma tumors was primarily based on the histopathological and microscopic features of hematoxylin and eosin stained sections. According to histological criteria, glial-derived tumors could be classified as either diffuse astrocytoma, anaplastic astrocytoma, oligodendrogliomas, anaplastic oligodendrogliomas, oligoastrocytomas, or glioblastoma [3]. These classifications provided important information about the cellular etiology and developmental differentiation of brain malignancies and were the primary means for grading tumors. However, accurate histological classification of brain tumors is not always straightforward. Although histological evaluation has remained the mainstay for brain tumor classification and grading for decades, there is increasing evidence that molecular genetic testing can supplement histology by identifying genetic biomarkers that are more common in some glioma subtypes than others. The utility of molecular genetic testing to aid tumor classification is best exemplified by the incorporation of several genotype biomarkers in the most recent WHO classification systems [2,4]. The most significant of these genetic biomarkers for the classification of adult-type diffuse gliomas include *IDH1* or *IDH2* mutation, 1p19q co-deletion, *TERT* promoter mutations, chromosome 7 gains with chromosome 10 losses, and *EGFR* amplification, as well as various other aberrations, i.e., *ATRX* alteration, *BRAF* alteration, *CDKN2A/B* homozygous deletion, *MYB* or *MYBL1* alteration, histone H3 mutation, and 10q23 (*PTEN*) losses [2]. Diffuse gliomas are the most prevalent form of gliomas in adults and are associated with a better overall prognosis. Molecular characterization of diffuse astrocytic and oligodendroglial tumors has led to the recognition of three molecular subtypes that better predict the survival of patients. The best survival is seen in Subtype 1, which is characterized by *IDH1* or *IDH2* mutation along with co-deletion of the chromosomal arms 1p and 19q. Intermediate survival is seen in Subtype 2, which is characterized by *IDH1/IDH2* mutation but intact 1p19q. The least favorable survival is seen in Subtype 3, which is characterized by normal *IDH1/IDH2* status and intact 1p19q [5,6]. The 2016 WHO CNS tumor classification included these molecular markers along with histological features to classify different subtypes of diffuse gliomas [2]. Three types of adult-type diffuse gliomas are also recognized in the most recent edition of the WHO CNS tumor classification (2021): astrocytoma, *IDH*-mutant; oligodendrogllioma, *IDH*-mutant and 1p19q-codeleted; and glioblastoma, *IDH*-wildtype [4].

Somatic *IDH1/IDH2* mutations occur in approximately 80% of gliomas and secondary glioblastomas but in less than 10% of primary glioblastomas [7]. Two hotspots for *IDH1/IDH2* mutation have been described in gliomas: *IDH1* codon 132 (p.R132) and *IDH2* codon 172 (p.R172). IDH1 and IDH2 normally catalyze the oxidative decarboxylation of isocitrate to alpha-ketoglutarate. Mutations at *IDH1* codon 132 and *IDH2* codon 172 enhance affinity for alpha-ketoglutarate (α-KG) which is then converted to D-2-hydroxyglutarate (D2HG), a competitive inhibitor of several α-KG-dependent dioxygenases (such as histone demethylases and the TET family of methylcytosine hydroxylases). The enzymatic inhibition induced by the accumulation of D2HG results in increased levels of histone H3 lysine methylation levels, global DNA hypermethylation (CpG island methylator phenotype, CIMP) and altered epigenetic regulation of cellular functions [8,9].

Immunohistochemistry (IHC) represents a common methodology for the direct detection of the most common IDH1 mutation in brain tumors, IDH1 p.R132H. However, a negative IHC result does not rule out the possibility of other *IDH1* or *IDH2* variants. Given the association between *IDH1/IDH2* mutations and CIMP, it has been demonstrated that methylation patterns may be useful as a surrogate marker for mutations of the *IDH1/IDH2*-associated epigenetic pathway, particularly in cases with less common *IDH1/IDH2* variants [8,10]. WGMA testing has the added potential for the concurrent detection of CIMP and relevant copy number variations (i.e., 1p19q co-deletion) in a single assay. In this study, we examined the use of whole-genome DNA methylation array (WGMA) testing to identify IDH1/IDH2 pathway disruption, through the detection of CIMP, in addition to the characterization of additional molecular alterations that may assist in the classification of glioma tumors. In addition, this study identified novel methylation loci associated with glioma CIMP that have the potential to refine the utility of WGMA assessment to detect CIMP and predict *IDH1/IDH2* mutation status.

## 2. Methods

### 2.1. Sample Selection

This study included a discovery cohort of 39 glioma tumors from the Hamilton region collected between 2010 and 2013. An additional replication cohort of 9 blinded glioma tissue DNA samples was included for WGMA testing. These samples were used to assess the feasibility of our methylation array analysis pipeline for assessment of the *IDH1/IDH2* mutation status in a blinded cohort. All samples from the discovery and replication cohort were anonymized and deidentified prior to testing. The study and use of tissues was reviewed and approved by the Hamilton Integrated Research Ethics Board.

Tumor histology was reviewed and interpreted by two senior neuropathologists as part of routine clinical care. Original tumor classifications were based on standard histopathology criteria regarding WHO grading at the time, including cellular morphology, mitotic activity, microvascular hyperplasia, and necrosis.

DNA was extracted from formalin-fixed paraffin-embedded (FFPE) tumor samples. Briefly, tissue curls from each specimen were deparaffinized using xylene and then washed with alcohol to enable DNA extraction using a QIAmp DNA Mini Kit (QIAGEN).

### 2.2. Immunohistochemistry for IDH1 R132H

Immunohistochemistry was performed on all retrospective samples using a standardized protocol and an anti-IDH1 R132H antibody. All immunohistochemistry results were reviewed by a senior neuropathologist.

### 2.3. 1p19q LOH Testing

Microsatellite markers on the short arm of chromosome 1 (D1S468 (1p36.3), 2D1S214 (1p36.31), D1S1161 (1p35.2)) and on the long arm of chromosome 19 (D19S559 (19q13.32), D19S412 (19q13.32), D19S601 (19q13.41)) were subjected to PCR amplification using fluorescently labeled primers and visualized by capillary electrophoresis. DNA from peripheral blood was compared with DNA extracted from tumor tissue. For informative markers, loss of heterozygosity (LOH) was calculated from the fluorescence capillary electrophoresis using the formula (B2/B1)/(T2/T), where B is the peak height of allele fragments amplified from blood and T is a peak height in tumor. Each marker is classified into one of the three categories: (1) LOH-positive (LOH value: ≥1.5 or ≤0.65), (2) inconclusive (LOH value: 0.66–0.79 or 1.21–1.49), or (3) no significant loss (LOH value: 0.80–1.20). 1p19q LOH was defined when at least 1 out of 3 markers in each arm (1p and 19q) were LOH-positive (Category 1) and there were no Category 3 markers.

### 2.4. Whole-Genome Methylation Array and CIMP Analysis

WGMA testing was performed on glioma tissue DNA of 48 individuals using the Infinium MethylationEPIC BeadChip (Illumina) according to a standard protocol. Briefly, FFPE DNA was bisulfite-treated using a Zymo Research EZ DNA Methylation Kit. Due to the degraded nature of FFPE DNA, the DNA was pre-processed using an Illumina Infinium HD FFPE Restore Kit, allowing for the samples to be suitable for amplification in the Illumina Infinium Methylation protocol. During the Infinium methylation protocol, samples underwent a whole-genome amplification, fragmentation, and purification prior to being hybridized to the EPIC array. Fluorescent probes hybridized to the array were detected using the Illumina iScan System. Sample quality was based on unmethylated and methylated median signal intensity as well as the number of probes with detection *p*-values above background (*p* < 0.05).

Beta values (estimate of methylation level using ratio of intensities between methylated and unmethylated alleles) were generated using Illumina Genome Studio Software and IDAT files were imported to Partek Genomic Suite software (PGS). An ANOVA test was performed using PGS to compare probe methylation levels between retrospective glioma samples with positive IHC for IDH1 R132H (13 individuals) and retrospective glioma samples with negative IHC for IDH1 R132H (26 individuals). We identified genomic regions (probes) with significant DNA methylation changes that met the following statistical criteria: (1) significant methylation change *p* < 0.01, (2) F-value (signal to noise) > 20, and (3) methylation estimate value (E, net methylation difference in IDH1-IHC-positive individuals as compared to IDH1-IHC-negative) > 15%. CpG probes with the most significant methylation changes were used to create a hierarchical clustering of differentially methylated genomic regions (epi-signature) in IDH1-p.R132H-positive versus IDH1-p.R132H-negative samples. Replication cohort samples were then plotted together in the hierarchical clustering of the epi-signature to classify their IDH status. A cutoff for average methylation beta values and Z-score was determined to classify CIMP status based on the selected CpG epi-signature and then applied to test results from each specimen within the replication cohort.

### 2.5. Copy Number Analysis

The 1p19q deletion analysis from the EPIC array data was performed according to published bioinformatics tools for copy number calling using the Conumee package [11]. Briefly, copy number calling consists of the following principal steps: normalization of probe intensities, calculation of the log R ratios (LRRs), and segmentation and determination of copy number status for each segment based on a chosen cut-off level or a *p*-value threshold. Baseline normalization was performed using blinded 1p19q non-deleted samples (previously tested by LOH). The cutoff for copy number determination was set at >±0.3.

### 2.6. Brain Tumor Methylation Classifier

IDAT files generated by Illumina Genome Studio Software were uploaded to v11b4 version of the online CNS tumor methylation classifier (https://www.molecularneuropathology.org (accessed on 27 July 2020)) and reports were produced as previously described [10].

### 2.7. Next-Generation Sequencing

Next-generation sequencing (NGS) was performed using Ion AmpliSeq™ Cancer Hotspot Panel v2, which targets hotspot regions of 50 cancer genes, including *IDH1* (NM_005896.2) and *IDH2* (NM_002168.2), on the Ion GeneStudio S5 sequencer (ThermoFisher, Waltham, MA, USA). Torrent Suite Software (version 5.4) was used for variant detection and Ion Reporter™Software (version 5.6, Waltham, MA, USA) was then used for the annotation of variants. Quality control parameters include a minimum coverage per base of 200×, and a minimum of 500× average coverage for *IDH1* and *IDH2*. All discovery and replication samples with sufficient DNA remaining were tested by NGS. This assay was used to test discovery samples with a discordant IDH status between IHC and DNA methylation array; and to confirm the IDH status of the replication samples after DNA methylation analysis. All samples met the quality control cutoff, except 1 sample (Sample 20) that had an *IDH2* coverage of 332×.

## 3. Results

### 3.1. Cohort Demographics

Tissues from thirty-nine tumors were included in the discovery cohort (refer to Appendix A). The average age of participants was 54 years (ranging from 27–82 years of age). Twenty-seven (69%) participants were male. Based on original histopathological classifications, 15 (38%), 14 (36%), and 10 (26%) samples were classified as oligodendroglioma (5 grade II, 10 grade III), oligoastrocytoma (7 grade II, 7 grade III), and glioblastoma, respectively.

The replication cohort included tissues from nine tumors (refer to Appendix A). The average age of participants in the replication cohort was 58 years (ranging from 32–71 years of age) and included four males (44%). Based on original histopathological classifications, five samples were classified as oligodendroglioma (two grade II, three grade III) and three were classified as glioblastoma. An original histopathological classification of the ninth sample was not available.

### 3.2. WGMA and CIMP

Quality control parameters: WGMA testing of the 48 samples yielded an average number of detected CpG (detection *p*-value < 0.05) of 846,023 (min = 746,229; max = 864,115), which demonstrates an adequate coverage of the total probes tested by the assay (over 850,000). For the selected 11 CpG sites (described below), the detection *p*-value was below 0.005 for all probes in all samples, except for one probe with *p* = 0.01 in one sample (refer to Appendix A).

The CIMP epi-signature was created using the discovery cohort of 39 tissue samples obtained from patients with diffuse glioma and was analyzed according to their IDH1 immunocytochemistry status. Various methylation changes at a single probe level were identified across the genome in the IDH1-IHC-positive group when compared to the IDH1-IHC-negative group. Using a less stringent cutoff of *p* < 0.01 and E > 10%, 851 CpG sites showed significant methylation change, the majority consisting of hypermethylation (844 CpG sites). A more stringent cutoff of *p* < 0.01, E > 15%, and F-value > 20 revealed 11 CpG sites, significantly hypermethylated, in the IDH-positive group (refer to Table 1 and Appendix A) compared to the IDH-negative group. The top 11 CpG sites comprise our epi-signature and were then used to create a hierarchical clustering of differentially methylated genomic regions, which demonstrated a unique methylation profile and sub-clustering of individuals positive for IDH1 p.R132H compared with individuals negative for IDH p.R132H (Figure 1). To identify the biological processes and molecular functions most enriched within our data set, we analyzed the 10 unique genes that overlapped with the 11 CpG sites that were differentially methylated in IDH-positive cases, using Database for Annotation, Visualization and Integrated Discovery (DAVID, https://david.ncifcrf.gov/ (accessed on 6 October 2022)). Due to a small number of genes, there were no significantly enriched categories after correction for multiple testing. However, interestingly, 5 out of 10 genes have roles in metal ion binding (*p*-value, 0.017, refer to Appendix A).

Comparison of the CIMP status by EPIC methylation array, the IDH1 mutation status tested by IHC and NGS is shown in Table 2. Initial classification of CIMP-positive versus CIMP-negative samples was based on hierarchical clustering using the 11 CpG sites. For the discovery cohort (*n* = 39), the average methylation (beta value; 0 = unmethylated; 1 = fully methylated) across those 11 CpG sites in CIMP-positive samples was significantly higher (mean = 0.63, SD = 0.25) than CIMP-negative samples (mean = 0.079, SD = 0.078) (refer to Appendix A). Three samples (Samples 21, 30, 33) with previous negative immunohistochemistry for IDH1 p.R132H were clustered in the IDH-positive group (Figure 1) and showed an average methylation of 0.60, 0.40, and 0.40. Sequencing analysis of two of those samples revealed a less common oncogenic *IDH1* variant c.394C>T (HGVS: NM_005896.3(IDH1):c.394C>T, p.(Arg132Cys)) in Sample 21, and an *IDH2* oncogenic variant, c.515G > A (HGVS: NM_002168.3(IDH2):c.515G>A, p.(Arg172Lys) in Sample 30. However, we could not obtain enough DNA to confirm the *IDH1*/*IDH2* mutation status of the third discordant sample (Sample 33).

One sample (Sample 9) was positive for an *IDH2* c.476G>A (HGVS: NM_002168.3(IDH2):c.476G>A, p.(Arg159His)) variant detected by NGS. This variant was interpreted as a variant of uncertain clinical significance (VUS) based on limited information in either public tumor and control population databases or in the available literature. Our CIMP analysis revealed that this sample clustered with the IDH-negative group and had an average methylation of 0.20, suggesting that this IDH2 variant does not significantly affect methylation status and is therefore likely benign.

Following the primary analysis with the discovery cohort, a small replication cohort (*n* = 9) with unknown *IDH1/IDH2* mutation status, was also examined (Figure 2). Analysis of additional replication cohort samples was not possible due to limited tissue and resources.

Hierarchical clustering analysis identified two samples (Sample 44 and Sample 45) that clustered with the hypermethylated IDH-positive group and seven samples that clustered with the IDH-negative group. The samples clustering with the IDH-positive group (CIMP-positive) had an average methylation beta value for the 11 CpG epi-signature sites of 0.42 and 0.75, whereas the remaining seven CIMP-negative samples had an average beta value ranging from 0.08 to 0.21. Subsequent sequencing analysis was able to confirm the presence of the common *IDH1* p.R132H (HGVS: NM_005896.3(IDH1):c.395G>A, p.(Arg132His)) mutation on Samples 44 and 45. All of the remaining samples were negative for either *IDH1* or *IDH2* oncogenic variants by NGS.

The hierarchical clustering data from the discovery and replication cohorts demonstrate suitable separation of IDH-positive versus IDH-negative samples and suggest that the top 11 epi-signature targets may be used to identify tumors with *IDH1* or *IDH2* methylation defects. Using a calculated Z-score from the average methylation from CIMP-negative samples (Appendix A), we observed that the majority of CIMP-positive samples (15 out of 17) have a score > 4 (average methylation value is 4 standard deviations higher than the CIMP-negative average), whereas CIMP-negative samples have a Z-score < 2. Using a stringent average methylation (beta value) cutoff of ≥0.4 and Z-score ≥ 4 for CIMP-positive prediction, and an average beta value ≤ 0.2 and Z-score ≤ 2 for CIMP-negative prediction, we could accurately predict the IDH status of 44 out of 48 samples. Analysis of additional samples is required to further refine these cutoffs and verify the sensitivity and specificity of this test.

### 3.3. 1p19q Co-Deletion Analysis

Because CNV analysis relies on the probe intensity measurement across all chromosomes, only samples with high overall intensity (unmethylated and methylated intensity (log2) > 9.5) were included in the CNV analysis. Using this quality cutoff, 18 samples were excluded from further copy number analysis (refer to Table 2; excluded samples are indicated by “—” in the “1p del”, “19q del”, and “additional CNV” columns). Methylation array data from the remaining 30 samples were evaluated for copy number variation and assessment of 1p19q co-deletion status.

For the previous 1p19q PCR testing, 12 samples had the 1p19q status classified as inconclusive (refer to Table 2; inconclusive previous 1p19q co-deletion results are highlighted). These inconclusive reports resulted from a lack of available testing, lack of informative markers in each chromosomal region, or segmental loss involving LOH in only one of the chromosomal arms. Thus, for concordance between methylation array and LOH testing, only samples that had LOH patterns consistent with 1p19q co-deletion or with no LOH detected (total 18) were included. For these 18 samples, LOH testing and DNA methylation array analysis were concordant for all 18 samples (Table 2). Five out of the eighteen samples were positive for 1p19q loss by LOH; DNA methylation array was able to confirm the loss in all five cases, although only weak 19q loss was noted in one sample (Sample 30). A plot for the normalized copy number analysis for one sample is represented in Figure 3A.

Copy number analysis of WGMA also demonstrated copy number variation in regions other than 1p and 19q (refer to Table 2) that may be relevant for tumor classification, including *EGFR* amplification (refer to Table 2 and Figure 3B). Further validation of the WGMA to reliably detect copy number variants of chromosomal segments other than 1p and 19q was beyond the scope of this project but are included in Table 2 for reference.

### 3.4. Brain Tumor Methylation Classifier

To compare the performance of the epi-signature against a pre-existing brain tumor methylation classifier system, methylation array data were assessed using a brain tumor methylation classifier developed by the German Cancer Consortium (DKTK) (https://www.molecularneuropathology.org (accessed on 27 July 2020)). The DKTK methylation classifier was chosen for analysis based on public availability at the time of the study and previous evidence demonstrating utility in the clinical setting [10,12,13]. Using the DKTK methylation classifier, the majority of our samples did not reach the optimal calibrated score threshold of ≥0.9 for methylation class and a threshold value of ≥0.5 for subclasses within methylation class families [10], probably due to suboptimal DNA quality (low intensity). Of those cases that reached the recommended scores (≥0.9 for methylation class; 0.5 for methylation subclasses; 15 samples), the IDH status and 1p19q co-deletion status were 100% concordant with our findings.

### 3.5. Refinement of Glioma Classification Using Genomic Information

According to the 2016 and 2021 World Health Organization Classification of Tumors of the Central Nervous System, molecular features should be incorporated in the definition of glioma tumors in addition to histological features [2,4,14]. Using the information obtained for *IDH* status and 1p19q co-deletion from WGMA, we could refine the classification of many tumor entities from the original histological classification (Table 2). In our cohort, a large number of oligoastrocytomas were observed in the histological classification. The 2016 WHO CNS classification strongly discourages the diagnosis of oligoastrocytoma, claiming that nearly all tumors with histological features suggesting both an astrocytic and an oligodendroglial component can be classified as either astrocytoma or oligodendroglioma using genetic testing for *IDH* mutation and 1p19q loss [2,4]. Following the current guideline, of the 14 oligoastrocytomas initially observed, 5 were reclassified to diffuse astrocytoma, *IDH* mutant; 5 were diffuse astrocytoma, *IDH*-wild-type; and 4 were oligodendrogliomas, *IDH* mutant, 1p19q co-deleted. In addition, our cohort seems to have an over-classification for oligodendrogliomas. Of the 20 initially classified oligodendrogliomas, only 4 met the genetic determinants for this entity (*IDH* mutation and 1p19q loss). Three histologically classified oligodendrogliomas had *IDH* mutation but intact 1p19q, being reclassified either to astrocytoma or diffuse astrocytoma, *IDH* mutant. Seven of the histologically classified oligodendrogliomas were reclassified as glioblastoma, *IDH*-wild-type. The remaining six samples designated as oligodendroglioma, NOS, were *IDH*-wild-type with 1p19q intact. However, careful examination of these genetic inconclusive specimens should be undertaken for features of glioblastoma or other tumors of similar histology.

## 4. Discussion

The advent of high throughput molecular techniques enabled the comprehensive characterization of genetic and epigenetic alterations in diffuse gliomas. Over the past few decades, several biomarkers of glioma have been identified, including *IDH1/IDH2* mutations, 1p19q co-deletion, *CDKN2A/B* homozygous deletion, chromosome 7 gains/chromosome 10 losses, *PTEN* deletion, monosomy 6, and genetic alterations in *ATRX*, *BRAF,* and *EGFR*, as well as promoter methylation of *MGMT* [8,15,16,17]. Many of these biomarkers are currently being clinically tested to aid in the diagnosis of glioma tumors as part of the revised 2021 WHO classification system of CNS tumors, as well as to provide a better prognosis and treatment choice for patients with these tumors. The current 2021 WHO classification system also introduced methylome profiling as an effective ancillary method for brain and spinal cord tumor classification when used alongside other, standard technologies, including histology.

In this study, we evaluated the use of a DNA methylation array to simultaneously identify the two most important biomarkers of glioma: *IDH1/IDH2* status and 1p19q co-deletion. It has been shown that *IDH1/IDH2* mutations are associated with a global change in the DNA methylation levels, known as a CpG-island methylator phenotype (CIMP), which can be used to characterize the *IDH1/IDH2* status of these tumors [8,12,18,19]. Initial studies using methylation data from the Illumina GoldenGate array had characterized CIMP in glioma tumors, which was associated with both glioma histological subtype and *IDH1/IDH2* mutations [8,19]. In 2017, Paul and colleagues developed a DNA methylation signature to identify glioma subtypes. Using different sets of CpG methylation biomarkers from the 450K array, Paul *et al*. were able to classify *IDH1/IDH2* mutant tumors versus *IDH1/IDH2* wild-type tumors, as well as oligodendroglioma, astrocytoma, and glioblastoma [18], proving to be a powerful tool for glioma classification. Using the EPIC methylation array, which is a more robust array covering > 850,000 CpG sites, and a discovery cohort of 39 glioma tissue samples, we successfully developed a DNA methylation (CIMP) signature to specifically distinguish *IDH1/IDH2*-positive samples from *IDH1/IDH2*-negative samples. This signature is composed of 11 CpG sites that were significantly hypermethylated in the *IDH*-positive group. It is unclear whether the hypermethylation of these 11 CpG sites directly impacts oncogenic processes. Intriguingly, 5 of the 10 corresponding genes have roles in metal ion binding, although the significance of this observation is uncertain. The 11 CpG sites were not included in previous published studies and may further refine the predictive value of methylation analysis to identify aberrant *IDH1/IDH2* pathways. Using this CIMP signature, we were able to correctly assign the *IDH1/IDH*2 status of an additional nine samples from the replication cohort. Although further studies with a larger number of replication cohort samples are required to fully validate this epi-signature, the preliminary findings from these samples suggest the utility of the signature to differentiate between *IDH1/IDH2* mutant and normal tissues. This represents an important finding that has the potential to refine the utility of WGMA to predict the *IDH1/IDH2* mutation status of gliomas, thus improving diagnostic yield and efficiency of laboratory testing compared to single analyte *IDH1/IDH2* tests. In addition to correctly identifying the IDH1/IDH2 status of tumors with the common IDH1 p.Arg132His variant, the reported epi-signature also enabled the identification of tumors with less common pathogenic *IDH1* and *IDH2* mutations, including one tumor with an IDH1 p.Arg132Cys variant and one tumor with an IDH2 p.Arg172Lys variant. The *IDH1/IDH2* mutation status would have been incorrectly assigned in these tumors if relying on immunohistochemistry for the common IDH1 variant as an isolated test. The CIMP signature may also provide additional information to better understand the functional consequences of *IDH1* or *IDH2* variants of unknown significance detected by NGS. As we presented here, NGS identified an *IDH2* variant (c.476G>A, p.(Arg159His)) that had been initially interpreted as a variant of uncertain clinical significance. The absence of significant hypermethylation at the examined loci and clustering of this variant with the *IDH*-negative group, suggests that the variant is unlikely to disrupt normal DNA-methylation-based epigenetic processes in this tumor and is therefore is likely benign. Nevertheless, it is important to note that CIMP is not a specific phenotype of *IDH1*/*IDH2* oncogenic variants but could also be detected in the presence of mutations in other regulators of the *IDH*-epigenetic pathway, such as *TET2* mutations in leukemia [20] and the *BRAF* p.V600E mutation in colorectal cancer [21,22]. Recent evidence has also identified a new group of *IDH*-positive brain tumors with hypomethylated genomes rather than genomic hypermethylation that is characteristic of most *IDH* mutant gliomas [23]. Although it has been suggested that the CIMP-specific DNA methylation targets may be non-overlapping and manifested by several molecular pathways across different tumor types [22], testing the CIMP status of glioma tumors only provides a subjective assessment of *IDH* mutation status.

In addition to *IDH* status, WGMAs can be used to evaluate chromosome copy number. Of particular interest herein are the chromosomes 1p and 19q, which are co-deleted in the oligodendrogliomas with concurrent *IDH1/IDH2* mutations. Using the R package Conumee for normalization and copy number calling, we detected 1p19q co-deletions as well as other relevant copy number variants, including chromosome 7 gains and chromosome 6 and 10 losses. The raw data also suggest possible detection of other clinically relevant molecular biomarkers, including *EGFR* amplification (refer to Figure 3B), *MET* (chromosome region 7q31.2) amplification, MDM2 (chromosome region 12q15) amplification, or *CDKN2A/B* (chromosome region 9p21.3) deletion. Detection of these molecular changes by WGMA can further streamline tumor classification by simultaneously screening for multiple relevant molecular markers in a single assay and limiting the number of tests required for tumors with limited sample quantity. Use of the EPIC WGMA for copy number assessment of these additional gene regions requires further validation against samples with a known copy number status of the relevant loci. An additional limitation of this study was the quality of the DNA extracted, presumably due to older tissue FFPE from our discovery samples (collected 5–8 years prior to WGMA testing and with either tumor heterogeneity or contamination with normal tissue). While copy number analysis was possible in all of the samples examined, poor overall WGMA array signal intensities in some samples made it clear that identification of subtle chromosome changes was more difficult than in other samples. More recently collected samples and microdissection of relevant tumor material in specimens with considerable tumor and normal tissue admixture may result in lower failure rates but we were unable to test this in the current study. Accordingly, the DNA quality was adequate and copy number calls were successful for the nine more recent (collected within 1 year) replication cohort samples that were included in the study, further highlighting the importance of specimen age as a pre-analytic consideration for FFPE tumor assessment using whole-genome methylation profiling.

Recently, a brain tumor methylation classifier was developed by the German Cancer Consortium (DKTK) to identify distinct DNA methylation classes of CNS tumors (free online tool: www.molecularneuropathology.org (accessed on 27 July 2020)) [10]. This classifier comprises 82 CNS tumor methylation classes and uses the information on methylation status (e.g., *IDH* status) as well as copy number changes (e.g., 1p19q co-deletion) to classify the tumors. In 2019, two studies evaluating the use of this classifier in glioma testing in a clinical laboratory showed that the classifier improved the diagnostic approach, especially for cases with ambiguous histology or non-informative molecular profiles [12,13]. However, these studies also highlight some limitations of this classifier in that a subset of samples did not reach the optimal calibrated score for use of this tool. Tumor cellularity and heterogeneity may represent additional limitations for the use of methylation-based classification tools. A study evaluating spatial heterogeneity showed that, while *IDH*-mutant gliomas have homogeneous methylation-based classification, intratumor variability in DNA methylation within glioblastomas, *IDH*-wild-type, and meningiomas can potentially affect tumor classification of these methylation-based biomarkers [24]. Applying the DKTK classifier to our study samples, only 15 out of 48 samples met the recommended calibrated scores to achieve optimal sensitivity and specificity for methylation class prediction [10]. Among these samples, *IDH1/IDH2* mutation status and 1p19 co-deletion were concordant with results obtained by our in-house protocol. However, for the remaining samples, the *IDH*1/IDH2 status was not reliably assigned, likely due to suboptimal quality of DNA extracted from FFPE material. Thus, our in-house protocol for EPIC DNA methylation analysis of *IDH1/IDH2* status and 1p19q co-deletion showed a better diagnostic yield than the other classifier used. Additional DNA methylation signatures for classification and/or prognostication of gliomas as well as other tumors have been developed in the last year and have been shown to be cost-effective and improve diagnostics over current clinical assays [25,26,27,28]. Further assessment of the utility of these classifiers for the detection of CIMP versus the CIMP epi-signature reported herein is beyond the scope of this project but represents an important area for future study.

The technical advantages of the EPIC DNA methylation array compared to IHC, targeted sequencing analysis, and copy number assay testing are grounded in three reasons: the first is the analytical advantage of using a single test to detect both *IDH* and 1p19q status, in addition to other copy number changes and promoter DNA methylation changes; the second is when there is a need to obtain a more accurate diagnosis for tumors with unusual or non-specific histology (i.e., oligoastrocytic tumors), and the third is when the molecular testing does not yield diagnostically informative results, such as in the case of *IDH1* VUS identification. Finally, the implementation of robust molecular techniques, such as EPIC array and next-generation sequencing, will allow for a more precise categorization of glioma tumors according to the 2021 WHO classification of CNS tumors, and ultimately improve the diagnosis and tailoring of patient therapy.

## 5. Conclusions

Accurate detection of *IDH1/IDH2* mutation and 1p19q co-deletion status is important for appropriate tumor classification of gliomas under the most recent 2021 WHO classification of central nervous tumors. We describe a novel DNA methylation epi-signature that specifically distinguishes brain tumors having oncogenic *IDH1/IDH2* variants from tumors with normal *IDH1/IDH2*. Copy number assessment of WGMA data further enabled accurate detection of tumors having 1p19q co-deletion. The ability to accurately detect *IDH1/IDH2* mutation status and 1p19q co-deletion in a single test enhances lab efficiency and reduces the need for additional testing of samples having limited material, Although further validation of this DNA methylation epi-signature is required, it has the potential to refine the utility of WGMA to predict *IDH1/IDH2* mutations status in gliomas, thus enhancing patient care through improved diagnostic yield and accuracy compared to single analyte *IDH1/IDH2* or 1p19q tests.

## Figures and Tables

**Figure 1 genes-13-02075-f001:**
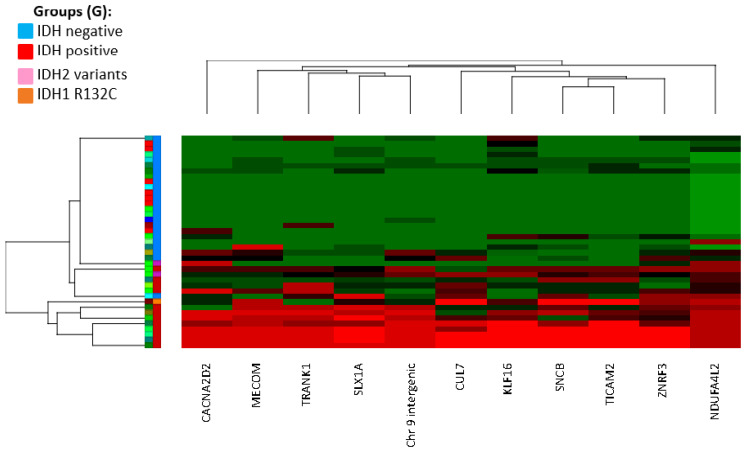
Hierarchical clustering of the top 11 CG sites (column) that were differentially methylated (epi-signature) in IDH-positive group versus IDH-negative group. Row; S: individual samples. G: group (blue: IDH-negative, red: IDH-positive, pink: *IDH2* variants by NGS, orange: *IDH1* variant by NGS).

**Figure 2 genes-13-02075-f002:**
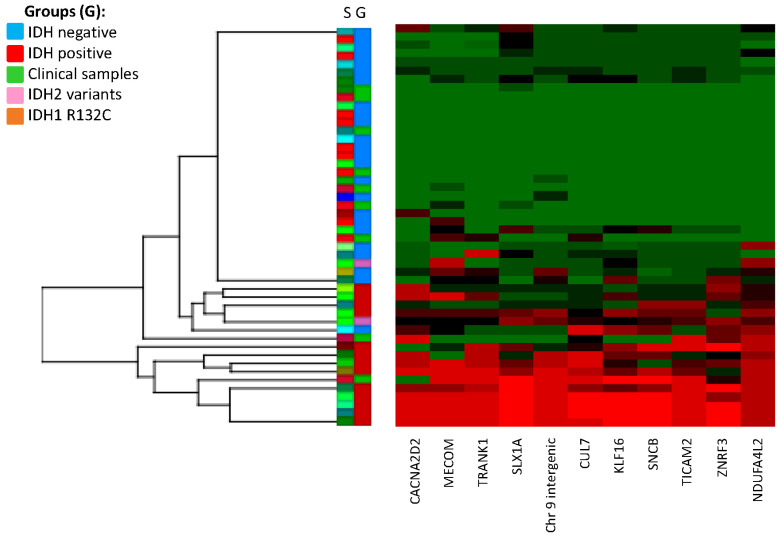
Hierarchical clustering of discovery and replication cohorts. The top 11 CG sites (column) that were differentially methylated (epi-signature) in IDH-positive versus IDH-negative group. Row; S: individual samples. G: group (blue: IDH-negative, red: IDH-positive, pink: *IDH2* variants by NGS, orange: *IDH1* variant by NGS, green: replication cohort samples).

**Figure 3 genes-13-02075-f003:**
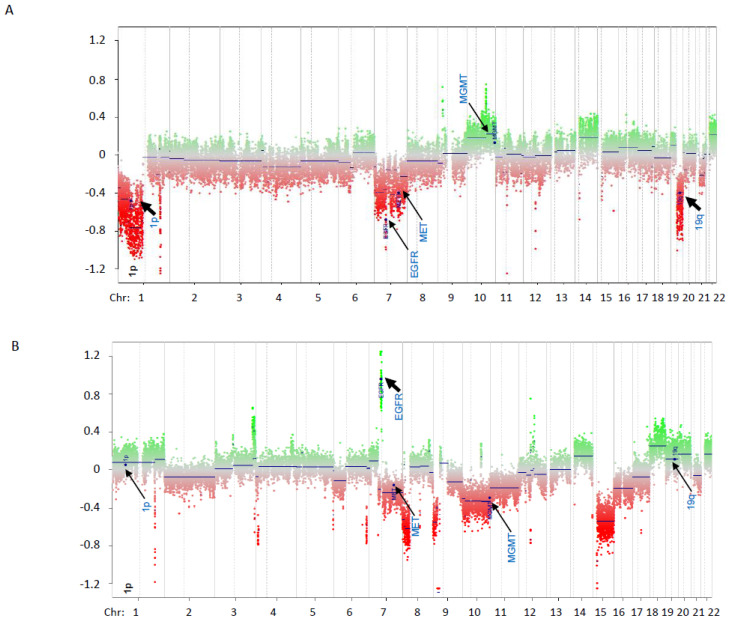
Genome-wide copy number analysis from methylation array data (Infinium MethylationEPIC BeadChip). All chromosomes are represented in the X-axis. Normalized copy number is represented in Y-axis (baseline = 0). Green dots represent probes with normalized copy number above 0 while red dots represent probes with normalized copy number below 0. Positions of probes for 1p, 19q, EGFR, MGMT and MET are indicated by arrows. (**A**) Example of genome-wide copy number analysis with 1p19q co-deletion (Sample 44). Instances of 1p19q co-deletion are highlighted by black arrows. (**B**) Example of genome-wide copy number analysis with EGFR amplification. EGFR amplification is highlighted by a black arrow.

**Table 1 genes-13-02075-t001:** CIMP epi-signature: top 11 CG sites that were differentially methylated in IDH-positive group versus IDH-negative group.

Probeset ID	Gene Symbol	Location within Gene	Genomic Location (hg19)	*p*-Value	Methylation Estimate Value	F Value
cg25042239	CUL7	5′UTR	Chr 6: 43021668	4.03E-06	15.2	27.6
cg12624667	TRANK1	TSS200	Chr 3: 36986555	9.81E-06	17	26.8
cg07003542	SNCB	5′UTR	Chr 5: 176056682	8.97E-06	16.7	26.1
cg13147822	MECOM	TSS1500	Chr 3: 168864313	2.19E-05	17.3	25.9
cg11952858	SLX1A	Body	Chr 16: 30206082	2.56E-05	15.5	25.6
cg13655082	intergenic	Not applicable	Chr 9: 125109046	1.37E-05	15.4	23.4
cg02870222	TICAM2	TSS200	Chr 5: 114938179	1.58E-04	17.3	22.7
cg13212283	NDUFA4L2	5′UTR	Chr 12: 57632120	1.19E-04	17.8	22.5
cg26325335	CACNA2D2	Body	Chr 3: 50402333	2.30E-04	16.2	22.4
cg03027661	ZNRF3	Body	Chr 22: 29426832	4.06E-05	15.7	21.1
cg15986590	KLF16	Body	Chr 19: 1862404	2.20E-04	15.5	20.5

**Table 2 genes-13-02075-t002:** Tumor reclassification based on histological, molecular, and epigenetic findings. Molecular testing results for each case are provided, including IDH1 p.R132H immunohistochemistry, *IDH1/IDH2* sequencing, CIMP positivity (as determined by hierarchical clustering, using the top 11 CpG sites that were differentially methylated) and 1p/19q status (based on single analyte testing and WGMA). Replication cohort samples are underlined. Tumor classifications as assessed using the MolecularNeuro-pathology.org Classifier (v11b4, accessed on 27 July 2020) are provided; calibrated scores for class and subclass are shown in brackets (recommended score class: >0.90, subclass > 0.5). Original tumor classifications as assessed by two senior neuropathologists are also provided. Histological criteria used for the original classifications include: oligodendroglioma grade II—no necrosis or microvascular hyperplasia (mvh); oligodendroglioma grade III—necrosis/mvh, increased mitotic activity; astrocytoma grade II—no necrosis/mvh or increased mitotic activity; astrocytoma grade III (anaplastic astrocytoma)—increased mitotic activity/anaplasia, no necrosis/mvh; glioblastoma—morphologically astrocytic with necrosis/mvh; oligoastrocytoma—morphologically mixed oligo/astrocytic, no necrosis/mvh or increased mitotic activity; oligoastrocytoma grade III—with anaplasia, increased mitotic activity.

Sample #	IHC Result (p.R132H)	NGS (IDH1/IDH2)	CIMP (Meth Array)	1p LOH	1p del (Meth Array)	19q LOH	19q del (Meth Array)	Other Apparent CNV (Meth Array)	Molcucular Pathology Classifier	Original Histopathology Classification	Reclassification as Per WHO 2021
1					--		--	--	plexus tumor, subclass pediatric B (0.5; 0.47)	Oligodendroglioma, grade II	Oligodendroglioma, IDH mutant, 1p19q co-deleted
2					--		--	--	plexus tumor, subclass pediatric B (0.48; 0.46)	Oligodendroglioma, grade II	Astrocytoma, IDH mutant
3								−7, −13, −18, +10	IDH glioma, subclass 1p/19q codeleted oligodendrioma (0.56; 0.47)	Oligodendroglioma, grade II	Oligodendroglioma, IDH mutant, 1p19q co-deleted
4								−7, −13, −18, +10, +22	plexus tumor, subclass pediatric B (0.31; 0.25)	Oligodendroglioma, grade II	Astrocytoma, IDH mutant
5								−7p partial, −7q, −9p, −11q partial, +14, +19, +22 partial	glioblastoma, IDH wildtype, subclass RTK I (0.99; 0.75)	Oligodendroglioma, grade III	Glioblastoma, IDH wild-type
6								−7, −13 partial, +14 partial, −18, −22	glioblastoma, IDH wildtype, subclass RTK I (0.48; 0.36)	Oligodendroglioma, grade III	Glioblastoma, IDH wild-type
7								+6q, −10, +13, −14 partial, +14 partial, +18, +20, −22,	glioblastoma, IDH wildtype, subclass RTK (0.99; 0.52)	Oligodendroglioma, grade III	Glioblastoma, IDH wild-type
8								+10, +13, +14 partial, −18, −20	glioblastoma, IDH wildtype, subclass mesenchymal (0,43; 0.36)	Oligodendroglioma, grade III	Glioblastoma, IDH wild-type
9		**			--		--	--	methylation class glioblastoma, IDH wildtype, subclass RTK I (0.51; 0.49)	Oligodendroglioma, grade III	Glioblastoma, IDH wild-type
10								+7, +9, +14, −18, −20, +22	methylation class glioblastoma, IDH wildtype, subclass RTK I (0.35; 0.2)	Oligodendroglioma, grade III	Glioblastoma, IDH wild-type
11								−1q, +2p,−2q, −3q partial, +4p, −7p, +8, +9p, +10p, −13, −18, +21, −21 partial, +22	methylation class glioblastoma, IDH wildtype, H3.3 G34 mutant (0.99; NA)	Oligodendroglioma, grade III	Glioblastoma, IDH wild-type
12								+7, −8p partial, −9p, +10, −13, +14, +20	No matching methylation classes with calibrated with score ≥ 0.3	Oligodendroglioma, grade III	Glioblastoma, IDH wild-type
13					--		--	--	methylation class plexus tumor, subclass pediatric B (0.56; 0.54)	Glioblastoma	Glioblastoma, IDH wild-type
14								query EGFR amp, −10p, +10q partial, −10q partial, +11p partial, +14, +19q, +22	methylation class glioblastoma, IDH wildtype, subclass RTK II (0.73; 0.47)	Glioblastoma	Glioblastoma, IDH wild-type
15								+7q, +19, +21	methylation class plexus tumor, subclass pediatric B (0.44; 0.4)	Glioblastoma	Glioblastoma, IDH wild-type
16								−1p partial, −6q partial, +7p, −7q, +9p partial, −13q partial, −14q partial, −15q partial, +16p, −16q, +22	No matching methylation classes with calibrated with score ≥ 0.3	Glioblastoma	Glioblastoma, IDH wild-type
17								+3, −4pter, +6p, −6q, query EGFR amp, +9q, +13, +14, −16, +19q, +22	methylation class glioblastoma, IDH wildtype, subclass RTK II (0.41; 0.17)	Glioblastoma	Glioblastoma, IDH wild-type
18								+6q, −7p, +10q, +13, +16q, +17, +19, +20	methylation class glioblastoma, IDH wildtype, subclass RTK II (0.82; 0.53)	Glioblastoma	Glioblastoma, IDH wild-type
19					--		--	--	methylation class plexus tumor, subclass pediatric B (0.51; 0.49)	Glioblastoma	Glioblastoma, IDH wild-type
20					--		--	--	methylation class plexus tumor, subclass pediatric B (0.52; 0.52)	Glioblastoma	Glioblastoma, IDH wild-type
21		***			--		--	--	methylation class family Plexus Tumor (0.42; 0.4)	Oligoastrocytoma, grade II	Astrocytoma, IDH mutant
22		--			--		--	--	methylation class plexus tumor, subclass pediatric B (0.45; 0.43)	Oligoastrocytoma, grade II	Astrocytoma, IDH mutant
23					--		--	--	methylation class IDH glioma, subclass 1p/19q codeleted oligodendroglioma (0.93; 0.9)	Oligoastrocytoma, grade II	Oligodendroglioma, IDH mutant, 1p19q co-deleted
24		--			--		--	--	methylation class plexus tumor, subclass pediatric B (0.5; 0.48)	Oligoastrocytoma, grade II	Oligodendroglioma, IDH mutant, 1p19q co-deleted
25									methylation class plexus tumor, subclass pediatric B (0.42; 0.4)	Oligoastrocytoma, grade II	Astrocytoma, IDH mutant
26					--		--	--	methylation class plexus tumor, subclass pediatric B (0.47; 0.45)	Oligoastrocytoma, grade III	Anaplastic astrocytoma, IDH-wildtype (WHO grade 3)
27					--		--	--	methylation class plexus tumor, subclass pediatric B (0.35; 0.32)	Oligoastrocytoma, grade III	Astrocytoma, IDH mutant
28								−7p, −9p, +10, −15, −18, +19p,	IDH glioma, subclass 1p/19q codeleted (0.9; 0.81)	Oligoastrocytoma, grade III	Oligodendroglioma, IDH mutant, 1p19q co-deleted
29								−4, −7, −9p, query +10, −12p, −12q partial, +22	methylation class IDH glioma, subclass 1p/19q codeleted oligodendroglioma (0.99; 0.98)	Oligoastrocytoma, grade III	Oligodendroglioma, IDH mutant, 1p19q co-deleted
30		****					*	−4, −18, query −9	No matching methylation clase s with calibrated score >= 0.3	Oligodendroglioma, grade II	Oligodendroglioma, IDH mutant, 1p19q co-deleted
31					--		--	--	No matching methylation clase s with calibrated score >= 0.3	Oligodendroglioma, grade II	Diffuse astrocytoma, IDH-wildtype (WHO grade 2)
32								−1q partial, −5p partial, −7, −9p partial, −13, +14, −18, +19, +22	methylation class glioblastoma, IDH wildtype, subclass RTK II (0.41; 0.2)	Oligodendroglioma, grade III	Anaplastic astrocytoma, IDH-wildtype (WHO grade 3)
33		--			--		--	--	methylation class plexus tumor, subclass pediatric B (0.39; 0.2)	Glioblastoma	Glioblastoma, NES
34					--		--	--	methylation class plexus tumor, subclass pediatric B (0.43; 0.41)	Oligoastrocytoma, grade II	Oligodendroglioma, IDH mutant, 1p19q co-deleted
35								+3, +9p partial, +9q, +12, +17	Plexus tumor, subclass pediatric B (0.49; 0.46)	Glioblastoma	Glioblastoma, IDH wild-type
36		--			--		--	--	methylation class plexus tumor, subclass pediatric B (0.42; 0.4)	Oligoastrocytoma, grade II	Diffuse astrocytoma, IDH-wildtype (WHO grade 2)
37					--		--	--	No matching methylation clase s with calibrated score >= 0.3	Oligoastrocytoma, grade III	Anaplastic astrocytoma, IDH-wildtype (WHO grade 3)
38								−1q partial, +1q partial, +6p, −6q, +10p, −13, query low level +16p, −16q, query low level 19q gain, +21, +22	Gliobastoma, IDH wildtype, subclass mesenchymal (0.98; 0.64)	Oligoastrocytoma, grade III	Anaplastic astrocytoma, IDH-wildtype (WHO grade 3)
39								−1p partial, +1p partial, +1q, −5q partial, −7q partial, −14q partial, +14q partial, −15q partial, +15q partial, −16p, +16q, +22	No matching methylation clase s with calibrated score >= 0.3	Oligoastrocytoma, grade III	Anaplastic astrocytoma, IDH-wildtype (WHO grade 3)
40				--		--		+2, +3q partial, +4q partial, +6q, +8, +10q partial, +14, +22	Glioblastoma, IDH methylation wildtype, subclass RTK II (0.99; 0.57)	Glioblastoma	Glioblastoma, IDH wild-type
41				--		--		−1p partial, −1q partial, +4q partial, +6q partial, −9p partial, −9q partial, 10q gain partial, +13, +14, +18q, +19p, +20, −21, +22	Glioblastoma, IDH wildtype, subclass mesenchymal (0.98; 0.94)	--	Glioblastoma, IDH wild-type
42								−2, +3q partial, −6p, +7p, EGFR amp, −7q, −8p, −9p partial−10, +10q partial, −11, +14, −15, −16, +18, +19, +20, +22	methylation class glioblastoma, IDH wildtype, subclass RTK I (0.99; 0.82)	Oligodendroglioma, grade III	Glioblastoma, IDH wild-type
43								−1p partial, −1q partial, −3p partial, −3q partial, −4q partial, −6q, −9p partial, −10 partial, −13 partial, −14, −18q, −22	methylation class glioblastoma, IDH wildtype, subclass RTK II (0.99; 0.81)	Glioblastoma	Glioblastoma, IDH wild-type
44	--							−7, +10, +14, +22	methylation class IDH glioma, subclass 1p/19q codeleted oligodendroglioma (0.99; 0.99)	Oligodendroglioma, grade II	Oligodendroglioma, IDH mutant, 1p19q co-deleted
45	--							−2, −3, −5, −7, +10, −11p, −12, −13, −15, −17, +18, +22	methylation class IDH glioma, subclass high grade astrocytoma (0.99; 0.91)	Oligodendroglioma, grade III	Astrocytoma, IDH mutant, grade 4
46	--							+3p partial, +3q partial, +4q partial, +6q, +9p partial, +10, −10q partial, +13q partial, +14, +18q, +22	methylation class glioblastoma, IDH wildtype, subclass RTK I (0.99; 0.95)	Glioblastoma	Glioblastoma, IDH wild-type
47	--							+6q, −7, +10, +13	methylation class control tissue, hemispheric cortex (0.95, NA)	Oligodendroglioma, grade II	Diffuse astrocytoma, IDH-wildtype (WHO grade 2)
48	--							−9p, −10q	methylation class glioblastoma, IDH wildtype, subclass RTK II (0.99; 0.83)	Oligodendroglioma, grade III	Glioblastoma, IDH wild-type

-- result not available; grey—inconclusive result; olive green—positive result; bule—samples with CNV analysis but inconclusive 1p/19q LOH; * low level 19q loss noted by methylation array; ** IDH2 variant of uncertain significance c.476G>A, p.Arg159His; *** IDH1 variant c.394C>T, p.Arg132Cys; **** IDH2 variant c.515G>A, P.Arg172Lys.

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
