# Peer review of "Evaluation of DNA Methylation Array for Glioma Tumor Profiling and Description of a Novel Epi-Signature to Distinguish IDH1/IDH2 Mutant and Wild-Type Tumors"

_genes, 2022, doi:10.3390/genes13112075_

Round 1

Reviewer 1 Report

In the current study, the authors investigate the utility of DNA methylation array for simultaneous detection of glioma biomarkers as a more effective testing strategy compared to existing single analyte tests. The authors declared they developed a DNA methylation signature to specifically distinguish IDH1/IDH2 mutant from normal samples.

I have some comments that should be addressed by the authors.

1. In the Abstract, the authors should reported how many samples were adopted for WGMA.

2. The authors should also reported the validation of their results for the DNA methylation signature.

3. In fact, only 39 samples were included in the discover stage and 9 were used for validation, the samples were too small for glioma. The experimental design lacks an appropriate power analysis. The authors should perform a calculation on the minimal sample size that provides sufficient statistical power to detect the required OR, knowing a priori the disease prevalence, minor allele frequency, the case/control ratio, and an expected error rate threshold.

4. I wonder if the cases were only for adult or children or both?

5. The authors declared DNA was extracted from formalin-fixed paraffin-embedded (FFPE) tumor samples.” The samples were collected between 2010 and 2013, the quality from FFPE samples may be low quality which may influence the results and accuracy.

6. The Figures were blurt and not clearly.

7. False-positive report probability and statistical power should be calculated for the positive findings for the samples included were so small.

Author Response

Reviewer #1

In the current study, the authors investigate the utility of DNA methylation array for simultaneous detection of glioma biomarkers as a more effective testing strategy compared to existing single analyte tests. The authors declared that they developed a DNA methylation signature to specifically distinguish IDH1/IDH2 mutant from normal samples. 

I have some comments that should be addressed by the authors.

  1. In the Abstract, the authors should reported how many samples were adopted for WGMA.

We agree with the reviewers comment and have corrected the abstract to include information about the number of samples tested by WGMA.

  1. The authors should also reported the validation of their results for the DNA methylation signature.

We recognize that one of the limitations of this study was the inability to fully validate the observed epigenetic signature using alternate technologies or a larger replication cohort.  Due to limited resources, it was not possible to increase the number of replication samples available for study. Nor was it possible to validate the observed epi-signature by a different method (ie. pyrosequencing or a different WGMA containing the 11 sites making up the signature).  Despite these limitations, we feel that the reported novel epi-signature represents an important preliminary finding that has the potential to refine the utility of WGMA to predict IDH1/IDH2 mutation status in gliomas, thus improving diagnostic yield and efficiency of laboratory testing compared to single analyte IDH1/IDH2 or 1p19q tests. 

The need for a larger sample size to validate the epi-signature was previously noted in the last sentence of Section 3.2 by the comment "Analysis of additional samples are required to further refine these cut offs and verify the sensitivity and specificity of this test”.  A sentence has been added to this manuscript section to clarify that analysis of a larger replication cohort was not possible due to limited tissue and resources.  Comments have also been added to the discussion to address this study limitation and highlight the need for future studies to fully validate the epi-signature, including expansion of the replication cohort size.  Validation of the epi-signature by a different technology may also be considered for future studies. 

  1. In fact, only 39 samples were included in the discover stage and 9 were used for validation, the samples were too small for glioma. The experimental design lacks an appropriate power analysis.  The authors should perform a calculation on the minimal sample size that provides sufficient statistical power to detect the required OR, knowing a priori the disease prevalence, minor allele frequency, the case/control ration and an expected error rate threshold.

Please see the response to question 2 that addresses the sample size of the replication cohort in the current study.  We agree with the reviewer that the sample size is too small, but feel strongly that these preliminary findings warrant publication based on their potential to refine the utility of WGMA to predict IDH1/IDH2 mutation status in gliomas. Additional comments have been added to the manuscript to acknowledge this study limitation.  Calculations for the minimal sample size to provide sufficient statistical power will be incorporated into future studies to fully validate this epi-signature. 

  1. I wonder if the cases were only for adult or children or both?

All of the cases examined were from adults.  In Section 3.1 “Cohort Demographics” the average age of participants for both the discovery and validation cohorts are stated.  For the discovery cohort: “ The average age of participants was 54 years (range 27-82 year of age)”.  For the replication cohort: “The average age of participants in the replication cohort was 58 years (range 32-71 years of age)”.

  1. The authors declared “DNA was extracted from formalin-fixed paraffin-embedded (FFPE) tumor samples.” The samples were collected between 2010 and 2013, the quality from FFPE samples may be low quality which may influence the results and accuracy.

We agree with this statement and have addressed limitations related to quality issues and the age of the FFPE samples in numerous places throughout the manuscript.  For example, in the result section ‘1p19q codeletion analysis’, we discuss removing 18 samples from the copy number variant assessment because of poor quality WGMA signal intensities.  In the result subsection ‘ Brain tumour methylation classifier’ we also discuss that several samples did not reach the optimal calibrated score threshold “probably due to suboptimal DNA quality (low intensity)”. The limitations of old FFPE samples on the observed quality of results is further discussed extensively In the fourth paragraph of the discussion section, (lines 80-92 of the original manuscript submission).

  1. The figures were blurt and not clearly.

Figures have been removed from the body of submitted revision and will be submitted as .pdf images instead.  We hope that this change in image format will improve quality of the figures. We defer to the editors; if there are images that are still blurry or unclear, please notify us so that the image quality can be addressed accordingly.

  1. False-positive report probability and statistical power should be calculated for the positive findings for the sample included were so small.

Thank you for this suggestion. We agree that these types of statistical measurements would be helpful and will be included in future studies to fully validate the epi-signature.  Limitations related to the size of the cohort in this preliminary report are addressed in the response to question 2.

Reviewer 2 Report

Schenkel, et al reports a new DNA methylation criterion to supplement current histological efforts to identify glioma subtypes specifically IDH1/2 and 1p19q status.  The investigators utilized whole genome methylation arrays on a total of 48 tumor samples.  The authors found that by conducting a whole genome wide methylation analysis, they were able to identify IDH1/2 wild-type/mutant, 1p and 19q loss, and developed an 11-GC epigenetic signature.  These eleven genomic sites were found to be hypermethylated in IDH1/2 positive samples.

The authors did a great job explicitly stating what their parameters were for the analyses.

Some manners of concern are:

1)    Of the 11- epigenetic signature sites, 10 sites are within known genes.  Do these genes have differential protein or mRNA expression levels between the IDH1/2 wildtype versus mutant?  The paper would greatly improve if the investigators showed a qPCR or a western blot of a few of these targets’ expressions and how it correlates with the methylation status.

2)    Additionally, how does the 11-epigenetic signature sites protein/mRNA expression correlate with IDH1/2 and 1p19q loss using not only the samples used in this paper but also looking at tumor expression databases such as R2, GLASS, etc?

3)    Is there a connection between the 10 genes from the epigenetic signature to any known dysfunctional cellular pathway in gliomas?  If the authors expand on how the methylation status of these genes impact on certain oncogenic pathways, it would greatly increase the impact of the signature and this paper.

4)    The authors have a figure 3B but does not discuss the importance of this figure within the Results section.  The paper would improve if the authors expand their analysis of using their WGMA and EGFR, MET, and MGMT statuses. 

5)    How does the new methodology predict patient outcomes/survival, therapy responsiveness, etc. as compared to the current criteria?

Author Response

Reviewer #2

Schenkel, et al reports a new DNA methylation criterion to supplement current hitological efforts to identify glioma subtypes specificall IDH1/2 and 1p19q status. The investigators utilized whole genome methylation arrays on a total of 48 tumor samples.  The authors found that by conducting a whole genome wide methylation analysis, they were able to identify IDH1/2 wild-type/mutant, 1p and 19q loss, and developed an 11-GC epigenetic signature.  These eleven genomic sites were found to be hypermethylated in IDH1/2 positive samples.

The authors did a great job explicitly stating what their parameters were for the analyses.

Some manners of concern are:

  • Of the 11- epigenetic signature sites, 10 sites are within known genes.  Do these genes have differential protein or mRNA expression levels between the IDH1/2 wildtype versus mutant?  The paper would greatly improve if the investigators showed a qPCR or a western blot of a few of these targets’ expressions and how it correlates with the methylation status.

We agree with the reviewer that correlation of DNA methylation with gene expression  would provide further confirmation of functional importance of the genes in oncogenic process in glioma. Unfortunately this was not possible for several reasons in our retrospective sample cohort 1) Most of tissue samples were exhausted after methylation and sequencing 2) Samples used in this study were collected in 2010-2013 and RNA/proteins are likely too degraded to obtain meaningful results. The advantage of studying of DNA methylation compared to gene expression that it is more stable than RNA over time.

  • Additionally, how does the 11-epigenetic signature sites protein/mRNA expression correlate with IDH1/2 and 1p19q loss using not only the samples used in this paper but also looking at tumor expression databases such as R2, GLASS, etc?

We have reviewed expression data from three studies PMID: 19920198, PMID: 16530701, PMID: 12670911 looking at gene expression in different sub-types of high grade gliomas cited by Glass consortium (PMID: 29432615). We did not find significant correlations with methylation targets presented in our paper and differentially expressed genes presented in the papers above. This could be at least in part explained by differences in gene coverage between platforms used in the gene expression studies and Epic methylation array used in this study. Due to absence of the meaningful correlation between gene expression and DNA methylation, this comparison was not presented in this paper. However, we agree with the reviewer that more extensive analysis of comparison of methylation and gene expression is warranted in the future. 

  • Is there a connection between the 10 genes from the epigenetic signature to any known dysfunctional cellular pathway in gliomas?  If the authors expand on how the methylation status of these genes impact on certain oncogenic pathways, it would greatly increase the impact of the signature and this paper.

Thank you for this suggestion.  We have examined the biological functions of the genes overlapping the 11 CpG sites differentially methylated in IDH1/IDH2 positive cases and have added appropriate text to the end of the Results – WGMA and CIMP subsection accordingly.  A comment has also been added to the discussion accordingly.

  • The authors have a figure 3B but does not discuss the importance of this figure with the Results section. The paper would improve if the authors expand their analysis of using their WGMA and EGFR, MET and MGMT statuses.

Figure 3B has been modified to show the copy number analysis of a sample with suspected EGFR amplification.  The figure is now referenced in the result and discussion sections. Discussion of the potential utility of WGMA to detect multiple molecular biomarkers in a single test has been added to the discussion section. However, this application of the test requires further validation and is beyond the scope of the current project.

  • How does the new methodology predict patient outcomes/survival, therapy responsiveness, etc. as compared to the current criteria?

Consistent with the research ethics board approval obtained for this study, all of the samples examined were anonymized and deidentified prior to testing.  As such, it was not possible to correlated the laboratory findings with these clinical outcomes. We recognize that patient outcomes/survival and therapy responsiveness are important factors to consider, such analyses are beyond the scope of this study. 

Reviewer 3 Report

The work by Schenkel and coworkers seeks a molecular signature of the DNA methylation profile of gliomas as a marker for diagnosis. The subject is extremely relevant due to the great heterogeneity within this group of tumors and the complexity of the chromosomal alterations that often occur. The classification of brain tumors only started to consider molecular markers from 2016, and even in the most recent classification, histological analysis is still necessary. Therefore, I suggest minor changes to the manuscript to clarify some points:

The authors cited and discussed the most recent classification of brain tumors (2021), however, in table 2 it seems to use the classification of 2016 (column "reclassification based on biomarkers). In the title of table 2, the authors claim to have reclassified the tumors based on molecular markers and histological findings. I suggest placing a column stating what histological findings were found on the slides (microvascular proliferation? Necrosis?).

Author Response

Reviewer #3

The work by Schenkel and coworkers seeks a molecular signature of the DNA methylation profile of gliomas as a marker for diagnosis. The subject is extremely relevant due to the great heterogeneity within this group of tumors and the complexity of the chromosomal alterations that often occur.  The classification of brain tumors only started to consider molecular markers from 2016, and even in the most recent classification, histological analysis is still necessary.  Therefore I suggest minor changes to the manuscript to clarify some points:

The authors cited and discuss the most recent classification of brain tumors (2021), however, in table 2 it seems to use the classification of 2016 (column “reclassification based on biomarkers). In the title of table 2, the authors claim to have reclassified the tumors based on molecular markers and histological findings.  I suggest placing a column stating what histological findings were found on the slides (microvascular proliferation? Necrosis?)

Thank you for the feedback regarding tumor classification.  The samples used in this study were originally classified by two senior neuropathologist based on histologic criteria and WHO classification systems at the time. A comment has been added to the methods section to clarify this point.  Histologic criteria used in the original classification (including mitotic activity, microvascular hyperplasia and necrosis) have been added to the table legend.  Tumor reclassification based on molecular biomarker has also been reexamined and relabeled in the table according to the revised 2021 WHO classifications.

Round 2

Reviewer 1 Report

Acceptable.